# Gal-2 Increases H3K4me^3^ and H3K9ac in Trophoblasts and Preeclampsia

**DOI:** 10.3390/biom12050707

**Published:** 2022-05-15

**Authors:** Laura Hahn, Sarah Meister, Mareike Mannewitz, Susanne Beyer, Stefanie Corradini, Uwe Hasbargen, Sven Mahner, Udo Jeschke, Thomas Kolben, Alexander Burges

**Affiliations:** 1Department of Obsterics and Gynecology, University Hospital, Ludwig-Maximilians-Universität Munich, Marchioninistr. 15, 81337 Munich, Germany; sarah.meister@med.uni-muenchen.de (S.M.); mareike.mannewitz@med.uni-muenchen.de (M.M.); susanne.beyer@med.uni-muenchen.de (S.B.); uwe.hasbargen@med.uni-muenchen.de (U.H.); sven.mahner@med.uni-muenchen.de (S.M.); udo.jeschke@med.uni-muenchen.de (U.J.); thomas.kolben@med.uni-muenchen.de (T.K.); alexander.burges@med.uni-muenchen.de (A.B.); 2Department of Radiation Oncology, University Hospital, Ludwig-Maximilians-Universität Munich, Marchioninistr. 15, 81337 Munich, Germany; stefanie.corradini@med.uni-muenchen.de; 3Department of Gynecology and Obsterics, University Hospital Augsburg, 86156 Augsburg, Germany

**Keywords:** preeclampsia, histone modification, H3K4me^3^, H3K9ac, galectin-2, syncytialisation, trophoblasts

## Abstract

Preeclampsia (PE) is a severe pregnancy disorder with a pathophysiology not yet completely understood and without curative therapy. The histone modifications H3K4me^3^ and H3K9ac, as well as galectin-2 (Gal-2), are known to be decreased in PE. To gain a better understanding of the development of PE, the influence of Gal-2 on histone modification in trophoblasts and in syncytialisation was investigated. Immunohistochemical stains of 13 PE and 13 control placentas were correlated, followed by cell culture experiments. An analysis of H3K4me^3^ and H3K9ac was conducted, as well as cell fusion staining with E-cadherin and β-catenin—both after incubation with Gal-2. The expression of H3K4me^3^ and H3K9ac correlated significantly with the expression of Gal-2. Furthermore, we detected an increase in H3K4me^3^ and H3K9ac after the addition of Gal-2 to BeWo/HVT cells. Moreover, there was increased fusion of HVT cells after incubation with Gal-2. Gal-2 is associated with the histone modifications H3K4me^3^ and H3K9ac in trophoblasts. Furthermore, syncytialisation increased after incubation with Gal-2. Therefore, we postulate that Gal-2 stimulates syncytialisation, possibly mediated by H3K4me^3^ and H3K9ac. Since Gal-2, as well as H3K4me^3^ and H3K9ac, are decreased in PE, the induction of Gal-2 might be a promising therapeutic target.

## 1. Introduction

With an incidence of 2–5% in Europe and the USA, preeclampsia is one of the most common pregnancy disorders [1,2]. This disease is responsible for 10–15% of all maternal deaths worldwide, making it a leading cause of maternal, perinatal and neonatal morbidity and mortality [3,4].

Preeclampsia is diagnosed when arterial hypertension exceeds 140/90 mmHg in association with at least one additional organ manifestation after the 20th week of pregnancy—for example, damage to the kidneys (proteinuria > 300 mg/24 h), the liver, the lungs or the central nervous system [1,2]. The time criterion of a manifestation after the 20th week of pregnancy was selected since a previous occurrence is indicative of pre-existing organ damage [1].

The specific pathophysiological cause of preeclampsia has not been fully clarified yet. However, increased apoptosis of extravillous trophoblasts (EVT) [5,6] and a dysfunctional placental immune response [7,8] appear to result in impaired trophoblast invasion into uterine tissue [9]. Impaired transformation of the uterine spiral arteries has also been shown to play a major role in the development of preeclampsia, being also associated with impaired trophoblast invasion and inflammatory immune regulations [2,10,11,12]. As a consequence, uteroplacental malperfusion occurs [9,13,14], which can lead to placental insufficiency and chronic ischemia [15], resulting in oxidative stress, hypoxic conditions and the release of anti-angiogenic factors, possibly associated with an impaired syncytialization, leading to local and systemic inflammatory reactions [10,11,16,17], all of which result in such typical symptoms as increased blood pressure and organ manifestations, e.g., proteinuria.

While there is currently no known causal therapeutic strategy for the treatment of preeclampsia [13], approaches with acetylsalicylic acid and antihypertensives exist [1,18,19]. Still, the only actual curative treatment for preeclampsia is the termination of the pregnancy and the removal of the placenta [3,18,20]. The lack of a specific therapy increases the importance of research to determine the cause of preeclampsia and discover targets for future therapy.

Long-term effects of preeclampsia for mother and fetus provide evidence of epigenetic changes in preeclampsia, including an increased risk of cardiovascular and metabolic disease, such as chronic ischemic heart disease, hypertension and type 2 diabetes [11,18,21,22,23]. In accordance with our previously published results showing that the transcription-promoting histone modifications H3K4me^3^ and H3K9ac are decreased in trophoblasts in preeclampsia [24], several groups have already shown epigenetic alterations during preeclampsia [11,16,22,25,26,27]. However, since the mere identification of the involved histone modifications does not offer insight into potential treatments, the investigation of influencing factors on these processes seems to be of more relevance to evaluate possible therapeutic options.

Galectins are part of the lectin family and bind to β-galactoside units [28,29,30,31,32]. They are generally upregulated in various pathological situations, such as inflammation and infection [31]. Furthermore, they show a wide intra- and extracellular expression [33], for example, in the placenta [28,29]. Galectin-2 (Gal-2) is known to be expressed by both human trophoblasts and trophoblast-cell-lines, such as BeWo cells [28].

The investigated Gal-2 is considered to be a stimulator of vascular growth and a regulator of metabolism [30,34]. It shows immunomodulatory effects by inducing apoptosis in T-cells and promoting macrophage differentiation [30,31,34,35,36]. Gal-2 also decreases the release of proinflammatory cytokines [36] and induces trophoblast invasion [30]. Several studies have already shown a reduced expression of Gal-2 in preeclampsia, both in peripheral maternal blood and in the placenta [30,34,37,38].

Up to now, there are no data available concerning the connection of histone modifications with Gal-2. A clear connection, however, was shown by the investigation of the expression of galectin-1 (Gal-1) after treatment of neoplastic cells with histone deacetylase (HDAC) inhibitors. HDAC inhibitors induce the expression of Gal-1 [39,40,41,42] and an overexpression of HDAC1 was shown to suppress expression of LGALS1, the gene coding for Gal-1 [43], while histone deacetylation leads to downregulation of the receptor of Gal-1 [44]. Furthermore, hyperacetylation of histones is known to lead to active transcription [15,16,25,26,45]. While histone modifications do not only affect the expression of Gal-1, there is also evidence for Gal-1 influencing histone modifications. Thus, Fan et al. [46] investigated the role of Gal-1 in hypertrophic cardiomyopathy. Their cell culture experiments demonstrated that overexpression of Gal-1 attenuated isoproterenol-induced hypertrophy of myocytes via inhibition of translation and reduction of phosphorylation of the histone deacetylase HDAC4. In addition, Maier et al. [42] demonstrated that histone deacetylase inhibition could only be induced in glioma cells showing strong Gal-1 activity, not in H4IIe-hepatoma-cells, which do not overexpress Gal-1.

Gal-1 is functionally similar to Gal-2. Both proteins are prototypical galectins that are symmetric homodimers [32,47,48]. Furthermore, they share 43% of their amino acid sequences, have similar carbohydrate binding sites and show partially similar staining profiles and similar immunological and metabolic effects [31,47,48,49].

Due to the similarities of Gal-1 and Gal-2 in their amino acid sequences and functions, an analogy may also be expected in the case of histone modifications. Therefore, the examination of a connection between Gal-2 and the histone modifications H3K4me^3^ and H3K9ac in preeclampsia could prove to be illuminating.

Different studies have shown reduced expression of Gal-2 in the placenta and an impaired syncytialisation during preeclampsia. As a result, we hypothesized that Gal-2 might influence syncytialisation through the regulation of H3K4me^3^ and H3K9ac.

Therefore, the aim of this study was to analyze the effect of Gal-2 on the histone modifications H3K4me^3^ and H3K9ac in BeWo and HVT cells as well as its influence on syncytialisation.

## 2. Materials and Methods

### 2.1. Sample for Statistical Analyses

To investigate possible relationships between the histone modifications H3K4me^3^ and H3K9ac and Gal-2, data on immunohistochemical staining of placentas impaired by preeclampsia from previously published research projects [24,30] were used. Each staining was scored separately with the immunoreactive score (IRS) in the decidua and the syncytium. Due to former research projects, the size of the investigated groups was predetermined, which is why we analyzed the IRS of 13 placentas of women with preeclampsia and 13 control placentas for statistical correlations. Further information as for example the weeks of gestation and maternal age of the used placentas is provide in the Appendix A. The analyzed tissue was collected at the University Hospital of the Ludwig-Maximilians University of Munich between 2007 and 2012.

### 2.2. Cell Culture

Cell culture experiments were performed to investigate the effect of Gal-2 on histone modification. For this reason, BeWo and HVT (human villous trophoblasts; ScienCell Research Laboratories, Carlsbad, CA, USA) cells were used. All cell culture experiments were performed as technical triplicates. Cells were cultured in RPMI-1640 + 10% FCS.

BeWo cells are originally derived from gestational chorionic carcinomas and are widely established as a trophoblast model cell line. Moreover, they are histologically classified as mitotic active cytotrophoblasts with a moderate percentage of syncytial differentiation and therefore are similar to the syncytiotrophoblast [50,51]. Furthermore, BeWo cells have similar expression patterns of trophoblast-specific antigens and various receptors as human trophoblasts [50,52].

HVT cells are a trophoblast cell line derived from human villous trophoblasts [53,54] also known as human cytotrophoblasts [55,56]. Cytotrophoblasts predominantly form the basal layer of the syncytium, but also migrate early in gestation to form EVTs in the decidua [16,18].

Per chamber of a four-well chamber slide, 50,000 BeWo or 50,000 HVT cells were seeded in 1 mL of growth medium (RPMI-1640 + 10% FCS). After the cells had adhered to the slide overnight, they were incubated for 32 h with recombinant human Gal-2 (Novus Biologicals, Littleton, CO, USA). Gal-2 was added in three different concentrations: 1 µg/mL, 0.1 µg/mL and 0.01 µg/mL. After the incubation, the cells were stained on the slide.

### 2.3. Immunocytochemical Staining

After Gal-2 incubation, all chamber slides were washed for 5 min with PBS and put in an ice-cold mixture of 50% methanol and 50% ethanol for 10 min. Afterwards, the slides were air-dried and washed with PBS and incubation with blocking solution followed (five minutes with ZytoChem Plus HRP Polymer System, mouse/rabbit—Zytomed Systems, Berlin, Germany—for the BeWo cells; 15 min with Ultra-V-Block—Thermo Fisher Scientific, Waltham, MA, USA—for the HVT cells). The primary antibodies (H3K4me^3^/H3K9ac/E-cadherin + β-Catenin; Table 1) were added and incubated for 16 h at 4 °C. Primary antibodies were diluted in PBS for the BeWo cells and in DAKO dilution medium (Agilent Technologies, Santa Clara, CA, USA) for the HVT cells. Since the procedure varies from this step for the BeWo and HVT cells, it is described separately for each. Between the single steps, the slides were washed with PBS.

#### 2.3.1. Staining Protocol for the BeWo Cells

The overnight incubation of the BeWo cells with the particular primary antibody was followed by a 20 min incubation with post-block reagent (ZytoChem Plus HRP Polymer System, mouse/rabbit; Zytomed Systems, Berlin, Germany) and a 30 min incubation with HRP polymer reagent (ZytoChem Plus HRP Polymer System, mouse/rabbit; Zytomed Systems, Berlin, Germany). Afterwards, chromogenic 3,3Diaminobenzidine (DAB+; Agilent Technologies; Santa Clara, CA, USA) reagent was added for 30 s before stopping the reaction in distilled water. For counterstaining, Mayer’s acidic haematoxylin was used for 90 s and cells were blued in tap water for 60 s afterwards. To cover the slides, the aqueous mounting medium Aquatex (Merck, Darmstadt, Germany) was used.

#### 2.3.2. Staining Protocol for the HVT Cells

After pretreatment and overnight incubation with primary antibodies (simultaneous primary antibodies, as for BeWo cells) a 30 min incubation with the secondary antibodies (Cy3-labeled goat-anti-rabbit IgG and Alexa Fluor 488-labeled goat-anti-mouse IgG; Table 1) at room temperature followed. Regarding the staining of histone modifications, only Cy3-labeled goat-anti-rabbit IgG was used. Slides were washed in between for 5 min with PBS.

After the incubation with the secondary antibodies, the cells were air-dried in the dark and covered with mounting medium containing DAPI to stain the nuclei for fluorescence.

### 2.4. Evaluation of Staining

Regarding the staining of the histone modifications H3K4me^3^ and H3K9ac in BeWo and HVT cells, a scoring system similar to the semi-quantitative immunoreactive score—which is widely used in immunohistochemical analysis—was used. This modified IRS was established by Kakkassery et al. [57]. Each slide was divided into four sectors which were scored separately from 0 to 3 (0 = no signal/not a single cell was stained; 1 = minor signal/cells were stained sporadically; 2 = moderate to high signal/a couple of cells were stained within a cell layer; 3 = high signal/a couple of cells were stained within different cell layers). The cell fusion staining was evaluated by intensity measurement using the software Zeiss ZEN (blue edition; Carl Zeiss AG, Oberkochen, Germany), since this staining is not restricted to the nucleus. For this purpose, both the raw data and the data averaged for the respective DAPI intensities were analyzed.

### 2.5. Statistical Analysis

For the statistical analysis, the software SPSS (version 26; IBM, Armonk, NY, USA) was used. Due to the lack of normally distributed variables, the non-parametric Mann–Whitney U test and the Kruskal–Wallis test—for differences in the mean values—and the Spearman’s Rho—for correlation analyses of H3K4me^3^ and H3K9ac and Gal-2—were used. Significance was assumed at *p* < 0.05. The correlation coefficient r was interpreted as follows: *r* < 0.3 weak correlation, *r* < 0.5 moderate correlation, *r* > 0.5 strong correlation. All values are given as means ± standard deviations.

## 3. Results

### 3.1. The Histone Modifications H3K4me^3^ and H3K9ac Correlate Significantly with Gal-2

Since both analyzed histone modifications (H3K4me^3^ and H3K9ac) and Gal-2 are significantly decreased in preeclampsia [24,30], the proteins were first examined for possible statistical correlations of their immunohistochemical staining results (Table 2).

Considering all samples in total, significant correlations between Gal-2 in the syncytium and H3K4me^3^ and H3K9ac are shown in both the syncytium (*r*_H3K4me3_ = 0.486, *p*_H3K4me3_ = 0.012; *r*_H3K9ac_ = 0.463, *p*_H3K9ac_ = 0.017) and the decidua (*r*_H3K4me3_ = 0.402, *p*_H3K4me3_ = 0.042; *r* = 0.559, *p*_H3K9ac_ = 0.003). In addition, decidual Gal-2 expression correlates with the methylated histone H3K4me^3^ in the syncytium (*r* = 0.450, *p* = 0.021) and the acetylated histone H3K9ac in the decidua (*r* = 0.545, *p* = 0.004).

When controls and placentas impaired by preeclampsia were examined separately, altered results were detected. While there were no significant correlations found between the histones and Gal-2 among the controls (Table 2), Gal-2 correlated significantly with the histone H3K9ac in placentas impaired by preeclampsia. Significant correlations were found between syncytial Gal-2 and H3K9ac in the syncytium (*r* = 0.556, *p* = 0.048) and decidua (*r* = 0.759, *p* = 0.003). In addition, Gal-2 correlated significantly with H3K9ac when the decidual expressions were considered (*r* = 0.752, *p* = 0.003).

### 3.2. Gal-2 Promotes the Histone Modifications H3K4me^3^ and H3K9ac in BeWo Cells

The immunocytochemical staining of BeWo cells showed an increased staining intensity and a higher number of stained cells after incubation with Gal-2 (Figure 1). 

The examination of the trimethylated histone H3K4me^3^ showed an increase in histone modification after incubation with Gal-2 (*p* = 0.001, Mann–Whitney U test). Moreover, the differences in H3K4me^3^ expression were found to be significant regardless of the added concentration of Gal-2 (*p* = 0.003, Kruskal–Wallis test). Furthermore, a significant correlation between the level of H3K4me^3^ and the amount of Gal-2 was found (*r* = 0.962, *p* < 0.001, Spearman’s Rho).

The investigation of the acetylated histone showed similar results. Both the comparison of the control versus incubation with Gal-2 (*p* = 0.004, Mann–Whitney U test), as well as the comparison of all groups with different amounts of Gal-2 (*p* = 0.003, Kruskal–Wallis test) showed significant differences. Additionally, the expression of H3K9ac was significantly positively correlated with the amount of Gal-2 used for incubation (*r* = 0.956, *p* < 0.001, Spearman’s Rho).

### 3.3. Gal-2 Promotes the Histone Modifications H3K4me^3^ and H3K9ac in HVT Cells

Based on the results of the BeWo cell culture, incubation with Gal-2 was repeated with HVT cells (Figure 2).

Regarding H3K4me^3^, a significant difference was found after incubation with Gal-2 compared to the untreated control group (*p* = 0.020, Mann–Whitney U test). Furthermore, there was a significant difference regarding all individual groups (*p* = 0.044, Kruskal–Wallis test) and a significant correlation between the amount of Gal-2 and histone modification (*r* = 0.735, *p* < 0.001, Spearman’s Rho).

The comparison of HVT cells incubated with and without Gal-2 revealed a non-significant difference regarding the expression of H3K9ac (*p* = 0.684, Mann–Whitney U test). In addition, no significant difference could be found between the individual groups (*p* = 0.900, Kruskal–Wallis test), nor was a correlation detected between Gal-2 and H3K9ac (*p* = 0.769, Spearman’s Rho). Since no significant results regarding H3K9ac could be found, a further descriptive analysis was conducted. H3K9ac showed increased expression after treatment with Gal-2 (1.52 ± 0.822) compared with the respective untreated control cells (1.25 ± 0.540). Furthermore, a slight elevation of H3K9ac in connection with the amount of added Gal-2 (0.01 µg/mL = 1.50 ± 1.021; 0.1 µg/mL = 1.50 ± 0.540; 1 µg/mL = 1.56 ± 1.068) was shown. 

### 3.4. Gal-2 Promotes Cell Fusion in HVT Cell Culture

Cell fusion staining of HVT cells was performed by the visualization of E-cadherin and β-catenin after incubation with Gal-2 (Figure 3). In this staining, cell fusion is characterized by reduced expression of E-cadherin and β-catenin [29]. Certain structural proteins, such as E-cadherin, are known to be decreased after cell fusion [58,59,60].

The staining of E-cadherin and β-catenin showed a significantly reduced intensity after incubation with Gal-2, both in the mean intensity (E-cadherin: control = 3226.19 ± 210.494, with Gal-2 = 2714.84 ± 228.327, *p* = 0.018; β-catenin: control = 2396.36 ± 370.594, with Gal-2 = 1897.21 ± 392.687, *p* = 0.145; Mann–Whitney U test) and in the mean intensity related to DAPI (E-cadherin: control = 8.79 ± 1.924, with Gal-2 = 5.86 ± 0.578, *p* = 0.009; β-catenin: control = 6.39 ± 0.389, with Gal-2 = 4.05 ± 0.717, *p* = 0.009; Mann–Whitney U test). 

Furthermore, an analysis of controls versus the individual concentrations of Gal-2 showed significant results as well. Even though the results for β-catenin were only significant when related to DAPI (*p* = 0.043, Kruskal–Wallis test), E-cadherin showed a significant difference in overall mean intensity (*p* = 0.030, Kruskal–Wallis test). 

## 4. Discussion

Preeclampsia is a serious pregnancy disorder associated with some of the highest rates of perinatal morbidity and mortality [3,4,61]. However, knowledge of the exact pathophysiology is still incomplete and, at present, only symptom-oriented therapy strategies exist [1,3,13,18,19,20,62]. In order to find a causal therapy for preeclampsia it is essential to determine the differences between healthy placentas and placentas affected by preeclampsia, at protein, cellular, genetic and epigenetic levels, and to investigate influencing factors.

In our present study we found a significant correlation between the histone modifications H3K4me^3^ and H3K9ac and Gal-2 in placentas impaired by preeclampsia. Due to the overall positive correlations of the histone modifications and Gal-2, the influence of Gal-2 on these histone modifications was investigated in two different trophoblast cell culture models. A significant increase in the histone modification H3K4me^3^ in BeWo and HVT cells cultured with Gal-2 and an increase depending on Gal-2 concentration were found. In contrast, H3K9ac was only significantly affected by Gal-2 in the BeWo cell culture, while merely a slight descriptive difference appeared in HVT cells between the control group and the groups incubated with Gal-2.

These differences depending on trophoblast cell type are of sufficient interest to be considered more in detail. While the underlying pathophysiological mechanism is not yet known, different characteristics of the cell lines allow possible conclusions to be drawn. Whilst HVT cells resemble cytotrophoblasts [55,56,63], BeWo cells have several characteristics of cytotrophoblasts but also functional similarities with the syncytiotrophoblast [51]; to a certain extent, they also exhibit fusiogenic properties [50,64].

To interpret our results properly, one has to consider the structure of the placenta and the different functions of various differentiated trophoblast types in the organ. The placenta connects fetal tissue to the uterine wall and exhibits tumor-like properties [65]. It develops from the fifth day after fertilization with the differentiation of trophoblasts, which are epithelial cells, with partially highly proliferative and invasive growth [18,66,67]. Starting at the time of implantation, the trophoblasts proliferate and fuse, resulting in the syncytiotrophoblast, which has no lateral cell borders, thus representing a single multinucleated cell [16,18,68]. During pregnancy, villous cytotrophoblasts continuously fuse with the syncytiotrophoblast and lose their proliferative activity [18]. The syncytiotrophoblast forms a boundary layer around the fetal tissue and controls maternofetal transfer [2,16,18]. Another part of the cytotrophoblast migrates into the decidua, leading to the formation of EVTs, which arrange themselves into cell columns and subsequently also lose their ability to proliferate [18].

Our immunohistochemical placenta staining showed a more prominent correlation of histone modifications and Gal-2 in the syncytium, which is consistent with the results of the cell culture experiments, showing an effect of Gal-2 on histone modifications in BeWo cells, exhibiting functional similarities with the syncytiotrophoblast. These results point to a significant role of Gal-2 possibly related to the analyzed histone modifications in the syncytium during preeclampsia. These findings are consistent with data found in the literature indicating diminished syncytialisation in preeclampsia [69]. Various acetylated histones, e.g., H3K9, H3K14 and H3K27, are known to be involved in impaired angiogenesis and syncytialisation in placental ischemia or hypoxia [25]. Moreover, H3K4me^3^ and H3K9ac are known to be related to syncytialisation. Since both histone modifications indicate active regions at gene enhancers and promoters [70,71], it could be hypothesized that they may be upregulated after successful syncytialisation, as was already demonstrated in BeWo cells [60].

Syncytiotrophoblast fusion occurs throughout pregnancy [18]. Gal-1, which is closely related to Gal-2 [31,48], leads to the stimulation of syncytium formation [29,38,72]. Therefore, to evaluate the influence of Gal-2 on histone modification more accurately, cell fusion and thus syncytialisation were examined in the present study, revealing increased cell fusion depending on the amount of Gal-2 added in culture.

Gal-2 is already known to be involved in placental formation and is expressed by cytotrophoblasts in the placenta of healthy women [37]. Furthermore, Gal-2 has an additional effect on placental angiogenesis [30,34] and it is known to be strongly reduced in the placentas of patients impaired by preeclampsia, as well as in the peripheral blood of women suffering from preeclampsia [30,37]. To investigate syncytialisation, we chose HVT cells, since BeWo cells themselves show fusiogenic potential and already resemble the syncytiotrophoblast. However, since we wanted to study the process of syncytialisation, which basically means the fusion of cytotrophoblasts to form the syncytiotrophoblast, we chose to study HVT cells which are cytotrophoblasts.

Since both H3K4me^3^ and H3K9ac are known to be activators of transcription [25,45] and have already been shown to be upregulated during syncytialisation [60,73], the results of the present study point towards an association of the histone modifications H3K4me^3^ and H3K9ac with increased cell fusion via promotion of transcription. According to our in vitro experiments, the examined histone modifications are affected by Gal-2. The decreased levels of Gal-2 in preeclampsia [30] could be partially responsible for the impaired histone modifications H3K4me^3^ and H3K9ac in preeclampsia [24]. Reduction of these histone modifications and Gal-2 might therefore result in a decreased syncytiotrophoblast fusion, leading to disturbed placentation [74]. Since our results showed a positive association between and a significant influence of Gal-2 on syncytialisation, our data suggest that this effect of the histone modifications may occur in interaction with Gal-2. These findings also reinforce the association between the expression of Gal-2 and preeclampsia which was postulated in previous studies [30].

Interestingly, our results indicate that H3K9ac may have a stronger influence on syncytialisation than H3K4me^3^, based on the theory that Gal-2 influences syncytialisation via histone modification. On the one hand, a stronger correlation is shown between H3K9ac and Gal-2 in the immunohistochemical staining, especially when only placentas impaired by preeclampsia are considered, whereas only a moderate correlation is shown between H3K4me^3^ and Gal-2. On the other hand, in the cell culture of the BeWo cells, a significant correlation with the added amount of Gal-2 was shown only with respect to H3K9ac.

Moreover, these results are in agreement with the findings of Li et al. [73], who demonstrated a specific influence of H3K9ac on syncytialisation. Assuming that H3K9ac indeed makes a greater contribution to syncytialisation, further research on whether syncytialisation can be influenced by histone deacetylases or inhibition might be worth considering.

Still, our work is not free of limitations. Even though both BeWo cells and HVT cells are recognized cell line models for the placenta, a culture of freshly isolated trophoblasts might be interesting, whereby a picture of the actual influence of Gal-2 on histone modification and syncytialisation could only be revealed in an animal model, since the further hormonal influences of the placenta are missing in a single trophoblast culture model. In addition, the different staining methods for the BeWo and HVT cells allow only limited comparability of the cell lines. Nevertheless, this choice of different staining methods was reasonable, even though immunohistochemical cell staining is more precise and less error-prone than immunofluorescence. By contrast, in the case of HVT cells we wanted to establish clear comparability with E-cadherin/β-Catenin double-staining in order to be able to draw conclusions about the influence of Gal-2 on syncytialisation via histone modification. Lastly, it remains to be noted that we have only demonstrated the respective correlations of Gal-2 with the histone modifications H3K9ac and H3K4me^3^ based on immunostaining at the protein level. Moreover, in our study we did not perform a sequencing analysis to verify the effects we showed at the gene level.

Regardless of the significance of the association between Gal-2 and the examined histone modifications, our results show an influence of Gal-2 on the histone modifications H3K4me^3^ and H3K9ac in trophoblasts, whereas the effect seems to be more dominant in the syncytiotrophoblast. Since Gal-2, as well as the histone modifications H3K4me^3^ and H3K9ac, are decreased in preeclampsia, a possible pathophysiological involvement in the development of preeclampsia can be assumed. While further in vivo research is needed to prove the association between Gal-2 and the histone modifications H3K4me^3^ and H3K9ac found in the present study, Gal-2 seems to be a promising therapeutic target in the treatment of preeclampsia.

## 5. Conclusions

In this study we were able to show a connection between Gal-2 and the histone modifications H3K4me^3^ and H3K9ac in trophoblast cells and preeclampsia. We postulate that Gal-2 stimulates syncytialisation, which might be mediated by H3K4me^3^ and H3K9ac. Since Gal-2, as well as H3K4me^3^ and H3K9ac, is decreased in PE, the induction of Gal-2 might be a promising therapeutic target in the treatment of preeclampsia.

## Figures and Tables

**Figure 1 biomolecules-12-00707-f001:**
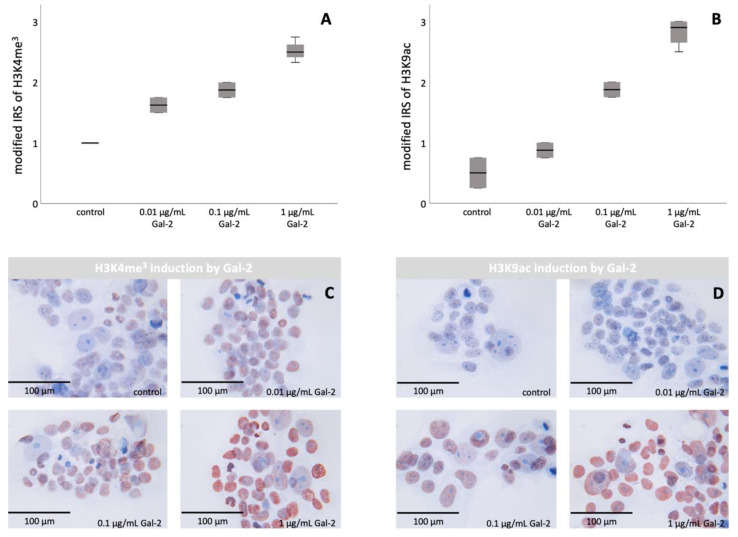
Induction of H3K4me^3^ and H3K9ac by increasing the concentration of Gal-2 in BeWo cells: staining results of immunocytochemistry of BeWo cells. (**A**,**B**) Representation of the staining data as boxplots. (**C**) Representative images of the staining of H3K4me^3^ after incubation with increasing Gal-2 concentrations. (**D**) Representative images of the staining of H3K9ac after incubation with increasing Gal-2 concentrations. Scale bar: 200 µm.

**Figure 2 biomolecules-12-00707-f002:**
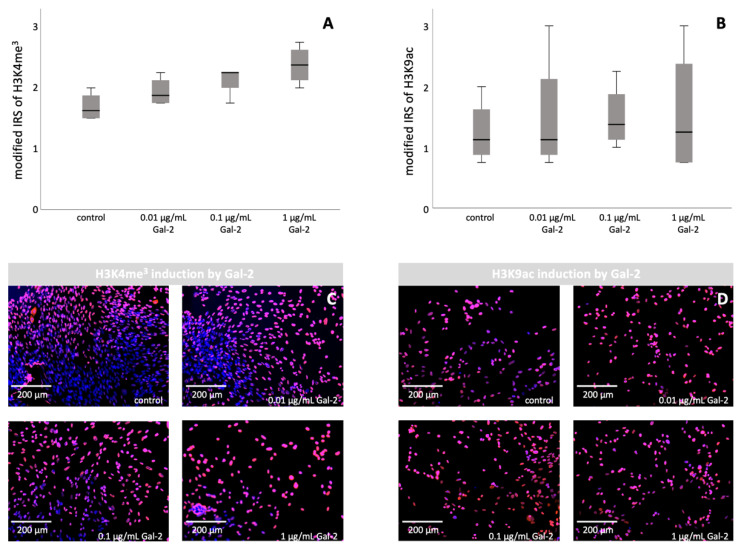
Induction of H3K4me^3^ and H3K9ac by increasing the concentration of Gal-2 in HVT cells: staining results of immunocytochemistry of HVT cells. (**A**,**B**) Representation of the staining data as boxplots. (**C**) Representative images of the staining of H3K4me^3^ after incubation with increasing Gal-2 concentrations. (**D**) Representative images of the staining of H3K9ac after incubation with increasing Gal-2 concentrations. Scale bar: 200 µm.

**Figure 3 biomolecules-12-00707-f003:**
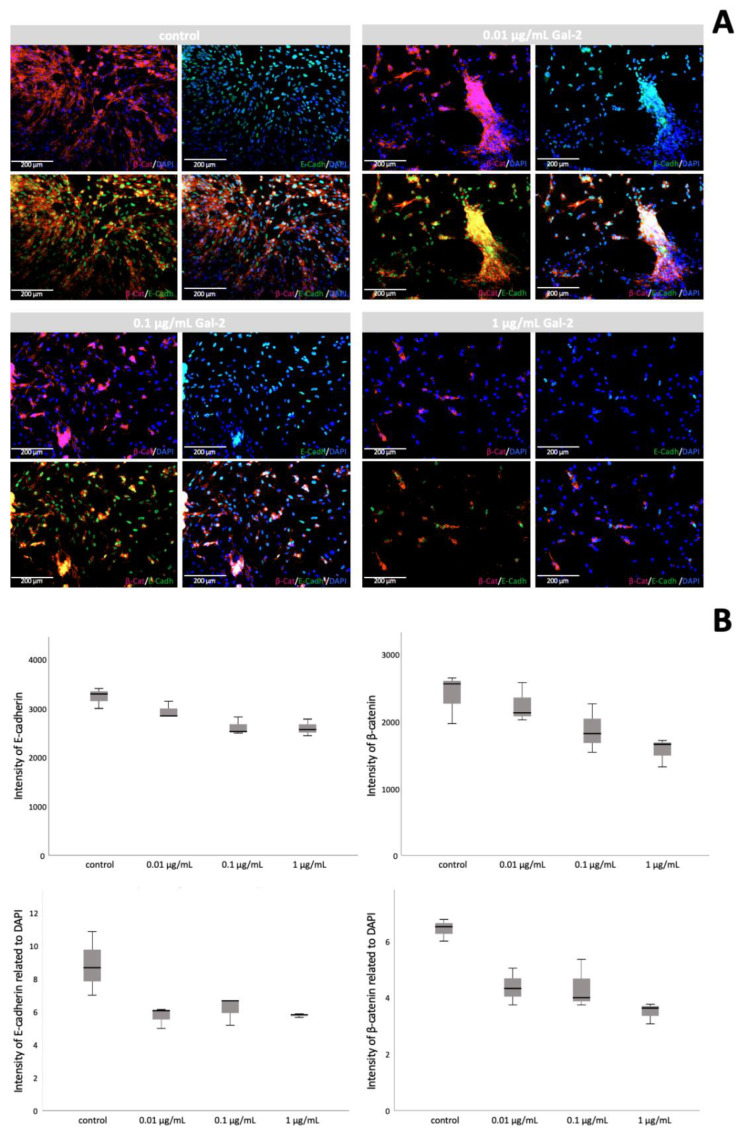
Induction of cell fusion by increasing the concentration of Gal-2 in HVT cells. (**A**) Exemplary images of cell fusion staining of HVT cells using β-catenin (red) and E-cadherin (green) after incubation with different concentrations of Gal-2. Nuclear staining with DAPI is shown in blue. Co-expression of β-catenin and E-cadherin results in yellowish staining. (**B**) Boxplots of the staining results of E-cadherin and β-catenin after incubation with increasing Gal-2 concentrations: mean intensity (upper row); mean intensity related to DAPI (bottom row).

**Table 1 biomolecules-12-00707-t001:** Antibodies used in immunocytochemistry of BeWo and HVT cells.

Antibody	Species Isotyp	Company	Dilution
H3K4me^3^	Rabbit IgG polyclonal	Abcam ab8580	1:100
H3K9ac	Rabbit IgG monoclonal, Y28	Abcam ab3129	1:200
Goat-Anti-Mouse Alexa Fluor 488	Goat IgG, polyclonal	Dianova 115-226-062	1:100
Goat-Anti-Rabbit Cy3	Goat IgG, polyclonal	Dianova 111-165-144	1:500
E-Cadherin	Mouse IgG1, monoclonal HECD-1	Calbiochem	1:50
β-Catenin	Rabbit IgG, polyclonal	Diagnostic BioSystems	1:50

**Table 2 biomolecules-12-00707-t002:** Correlations of the histones with other proteins. The upper value is the correlation coefficient r; the second value is the *p*-value.

		Gal-2 in All Placentas		Gal-2 Only in Controls		Gal-2 Only in PE	
		Syn	Dec	Syn	Dec	Syn	Dec
H3K4me^3^	Syn	0.486 * 0.012	0.450 * 0.021	0.390 0.188	0.366 0.219	0.247 0.416	0.285 0.345
	Dec	0.402 * 0.042	0.286 0.156	0.193 0.527	0.234 0.441	0.322 0.283	0.126 0.682
H3K9ac	Syn	0.463 * 0.017	0.368 0.065	0.019 0.951	0.020 0.948	0.556 * 0.048	0.456 0.117
	Dec	0.559 ** 0.003	0.545 ** 0.004	0.197 0.519	0.151 0.621	0.759 ** 0.003	0.752 ** 0.003

Correlation analysis was performed with the Spearman’s Rho: * significant (*p* < 0.05); ** highly significant (*p* < 0.01). Dec, decidua; Syn, syncytium.

## Data Availability

The datasets generated during the current study are available from the corresponding author on reasonable request.

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
