# Peer review of "Gal-2 Increases H3K4me^3^ and H3K9ac in Trophoblasts and Preeclampsia"

_biomolecules, 2022, doi:10.3390/biom12050707_

Round 1

Reviewer 1 Report

In the present work, Hahn Laura1 et al. try to explain that Gal-2 increases H3K4me3 and H3K9ac in trophoblasts and preeclampsia. There are some questions that should be explained.

  1. Lines 37-59, there are more paragraphs to explain PE. These contents should been refined.
  2. Line 93, Maier, Legare and Hanneman should been changed to Maier et al.
  3. Line 101, the objective and hypothesis should been added.
  4. Line 112, please provide the detail information for the placentas of women with preeclampsia and control placentas.
  5. Ethics approval is by the Institutional Ethics Committee of the LMU Munich (protocol code 18-700, date of approval 25.03.2019). However, the analyzed tissue was collected at the University Hospital of the Ludwig-Maximilians University of Munich between 2007 and 2012. Is these suitable?
  6. Lines 124, 128, there are too many references to explain syncytiotrophoblast (50-54) and human cytotrophoblasts (58-60).
  7. Figure 1, the BeWo-cells are too small. Please enlarge the BeWo-cells.
  8. Line 380, a conclusion section may be needed.
  9. Lines 420, 432, 434, 450, 452, 487, 489, 500, 543 and 547, please use capital letters only for the first letter in the title.

Author Response

In the present work, Hahn Laura1 et al. try to explain that Gal-2 increases H3K4me3 and H3K9ac in trophoblasts and preeclampsia. There are some questions that should be explained.

1. Lines 37-59, there are more paragraphs to explain PE. These contents should been refined.

We tried to go into more detail while explaining the different components of the development of PE and refined the content. We hope the reviwer is satisfied with the improved version of the introduction.

2. Line 93, Maier, Legare and Hanneman should been changed to Maier et al.

Thank you very much for this comment, we changed it in our manuscript.

3. Line 101, the objective and hypothesis should been added.

We agree with you that objective and hypothesis need to be added and adjusted our manuscript.

4. Line 112, please provide the detail information for the placentas of women with preeclampsia and control placentas.

We provided the detailed information of the placentas of women with preeclampsia which were available in Supp. Table 1.

5. Ethics approval is by the Institutional Ethics Committee of the LMU Munich (protocol code 18-700, date of approval 25.03.2019). However, the analyzed tissue was collected at the University Hospital of the Ludwig-Maximilians University of Munich between 2007 and 2012. Is these suitable?

We apologize for this mistake we added the correct ethics approval for the tissue collection in the method section.

6. Lines 124, 128, there are too many references to explain syncytiotrophoblast (50-54) and human cytotrophoblasts (58-60).

Thank you for pointing this out to us, we deleted several references which explain the trophoblast subtypes and concentrated on the most important ones.

7. Figure 1, the BeWo-cells are too small. Please enlarge the BeWo-cells.

We adjusted Figure 1 and enlarged the BeWo-cell pictures in the figure. We hope you find the new version of Fig. 1 more appropriate.

8. Line 380, a conclusion section may be needed.

Thank you very much for that comment we included a conclusion section.

9. Lines 420, 432, 434, 450, 452, 487, 489, 500, 543 and 547, please use capital letters only for the first letter in the title.
We adjusted the references and used capital letters only for the first word in the title.

For the Table and the Figure, please see the attachment.

Reviewer 2 Report

Dear authors! This article on the pathogenesis of preeclampsia. The article shows the results of various studies both on cell cultures and on BeWo-cells cell culture, as well as cells obtained from patients with preeclampsia. The Materials and Methods section may be written in more detail.

The article presents an original approach to the influence of Gal -2 in preeclampsia. After minor additions and changes, the article can be published in the journal.

Author Response

Dear authors! This article on the pathogenesis of preeclampsia. The article shows the results of various studies both on cell cultures and on BeWo-cells cell culture, as well as cells obtained from patients with preeclampsia. 

The Materials and Methods section may be written in more detail.

We tried to adjust our Material and Methods section to explain our methods more in detail, we hope we could meet the requirements of the reviewer.

The article presents an original approach to the influence of Gal -2 in preeclampsia. After minor additions and changes, the article can be published in the journal.

Thank you very much for your comments and for appreciating our work. We are very pleased that you find our manuscript appropriate to be published in the journal after the changes.

Reviewer 3 Report

Dear Authors.

The manuscript presented by you is of excellent quality, very good introduction. The methodology is well described and cited, however, when describing the statistics, it would be very clarifying to mention in each of the tests used, which data would be analyzed by each statistical test. In the results section, I have some comments, but in general the main findings are well described, however, in Table two, I fail to understand what is the importance of the correlation analysis of all the data together (first column on the left, titled Gal-2 in total).  Also the abbreviations used have not been presented anywhere in the manuscript, so they must clarify what Syn and Dez stand for. Figures 1, 2 and 3 in the graphs should say induction by Gal-2, although it is in the images of the cultures, each panel or figure should be self-explanatory, and it helps to read quickly and more comprehensively. Also, please add to the figure captions a sentence explaining that they are induced by increasing concentrations of Gal-2. The discussion section was well done.

Author Response

Dear Authors.

The manuscript presented by you is of excellent quality, very good introduction. The methodology is well described and cited, however, when describing the statistics, it would be very clarifying to mention in each of the tests used, which data would be analyzed by each statistical test.

Thank you very much for appreciating our data and our manuscript. We added more details about which test was used for which data generation, either in the method section and the result section.

In the results section, I have some comments, but in general the main findings are well described, however, in Table two, I fail to understand what is the importance of the correlation analysis of all the data together (first column on the left, titled Gal-2 in total).  

Thank you for your comment. In this column, we had a look at the correlation between the expression of Gal-2 and the respective histone modifications in the syncytium respectively the decidua in the whole collective, meaning in all placentas. In this first data analysis, we wanted to check whether there is a correlation at all. Moreover, this data analysis makes sense for us, because we are limited by our small collective when dividing our collective according to preeclampsia and control placentas and thus can only assess the correlation to a limited extent. The analysis of the whole collective is nevertheless meaningful - even though one notices in the analysis of the individual groups that the correlation seems to be predominant especially in preeclampsia - since in a correlation only the values of each placenta are compared with itself.

To avoid irritation, we have therefore decided to rename the column from "in total" to "in all placentas".

Also the abbreviations used have not been presented anywhere in the manuscript, so they must clarify what Syn and Dez stand for.

Thank you very much for this comment we added the abbreviations in the table legend and changed “Dez” to “Dec”.

Figures 1, 2 and 3 in the graphs should say induction by Gal-2, although it is in the images of the cultures, each panel or figure should be self-explanatory, and it helps to read quickly and more comprehensively.

We totally agree with you and changed the figures and hope they are now more self-explanatory now. Regarding Figure 3, we thought the figure itself was already self-explanatory, however we can add an "induction by Gal-2" to the different concentrations in the grey bars if you wish so.

Also, please add to the figure captions a sentence explaining that they are induced by increasing concentrations of Gal-2. The discussion section was well done.

Thank you very much for this addition, we supplemented corresponding sentence in the figure captions.

For the figures, please see the attachment.

Round 2

Reviewer 1 Report

We thank the authors for responding well to the modifications noted. The paper is worthy of considered publication after a minor revision.

Please put the hypothesis first, and then the objective.

Author Response

We thank the authors for responding well to the modifications noted. The paper is worthy of considered publication after a minor revision.

Please put the hypothesis first, and then the objective.

Thank you very much for your comment and for appreciating our work. We assume that you meant the paragraph in the introduction and hope that we have changed the manuscript according to your wishes.
